# Pet Reptiles—Are We Meeting Their Needs?

**DOI:** 10.3390/ani11102964

**Published:** 2021-10-14

**Authors:** Alexandre Azevedo, Leonor Guimarães, Joel Ferraz, Martin Whiting, Manuel Magalhães-Sant’Ana

**Affiliations:** 1CIISA—Centro de Investigação Interdisciplinar em Sanidade Animal, Faculdade de Medicina Veterinária, Universidade de Lisboa, 1300-477 Lisbon, Portugal; martinwhiting@mac.com (M.W.); mdsantana@fmv.ulisboa.pt (M.M.-S.); 2Instituto de Ciências Biomédicas Abel Salazar, Universidade do Porto, Rua Jorge de Viterbo Ferreira 228, 4050-313 Porto, Portugal; 3CIVG—Vasco da Gama Research Centre, Department of Veterinary Sciences, Vasco da Gama University School, Avenida José R. Sousa Fernandes 197, 3020-210 Lordemão, Portugal; mleonor93@gmail.com; 4Exoclinic—Clínica Veterinária, Rua D. António Ribeiro, Nº1, Loja B, 1495-049 Algés, Portugal; 5Centro Veterinário de Exóticos do Porto, Rua Arquitecto João Andresen 132, 4250-242 Porto, Portugal; joeltferraz@hotmail.com

**Keywords:** companion animals, animal welfare, exotics, controlled deprivation, reptiles, behavior, snakes, lizards, chelonians

## Abstract

**Simple Summary:**

Little is known about the ability of reptile owners to meet the needs of their pet animals. In this study, 220 pet reptile owners in Portugal reported their knowledge of reptile behaviors and the provision of essential husbandry needs (temperature, lighting, diet and refuge). Although two-thirds of respondents scored very good to excellent in terms of interpreting their pet reptile’s behaviors, 85% failed to provide at least one of the four husbandry needs. Moreover, behaviors indicative of poor welfare and captivity stress were considered ‘normal’ by some respondents. These results suggest that many pet reptiles in Portugal live in, at best, ‘controlled deprivation’ and are at risk of suffering poor welfare throughout their lives. Despite this, none of the respondents reported their reptile’s welfare as very poor, and a single respondent reported it as poor. We suggest that poor welfare and abnormal behaviors in pet reptiles have become accepted as normal, precluding the search for ways to prevent them. These results suggest that campaigns aimed at challenging the current norm for adequate reptile welfare are warranted. In particular, the predominant view, propelled by the exotic pet industry, that reptiles are low-maintenance pets needs to be actively refuted.

**Abstract:**

The ability to meet the needs of each species in captivity is at the heart of the ethical debate on the acceptability of keeping reptiles and other animals as pets. Little is known about the ability of reptile owners to understand their pets’ behavior and to meet their welfare requirements. In this study, we surveyed pet reptile owners in Portugal (N = 220) to assess their behavioral knowledge and the provision of essential husbandry needs. Although two-thirds of respondents (68%) scored very good to excellent in terms of knowledge of their pet reptile’s behaviors, only 15% of respondents met four essential reptile husbandry needs (temperature, lighting, diet and refuge) and 43% met two or less. None of the respondents reported their reptile’s welfare as very poor, and only a single respondent reported it as poor. Logistic regression model showed that while snake owners had fourteen times higher odds of reporting adequate husbandry provision, lizard owners had the highest odds of reporting good or very good welfare despite providing less of their animals’ basic husbandry needs. These results suggest that many pet reptiles in Portugal live in, at best, ‘controlled deprivation’ and are at risk of suffering poor welfare throughout their captive lives. Moreover, behaviors indicative of poor welfare and captivity stress were considered ‘*normal*’ by up to one quarter of respondents. We suggest that the frequency of these behaviors in pet reptiles has led to their acceptance as normal, precluding the search for ways to prevent them. These results suggest that campaigns aimed at challenging the current norm for adequate reptile welfare are warranted.

## 1. Introduction

The ability to meet the needs of each species in captivity is at the heart of the ethical debate on the acceptability of keeping exotic pets [1,2,3]. While some authors question whether there is evidence to support a disproportionately high incidence of welfare problems in reptiles compared to other species [3], others suggest that reptile needs cannot be easily met in captivity [1,4]. Consequently, there has been an increasing interest in the assessment of the welfare of captive reptiles [5,6] and of the major barriers pet owners [1,4] and veterinarians [7] face in meeting their needs. 

Conservative estimates indicate that approximately 8 million reptiles are kept as pets in the European Union [8]. However, the number of animals traded to allow this number of pets may exceed 24 million, given the reported first-year mortality estimates in homes that range from 3.6% [9] to 75% [10]. The large range of these estimates reflects an ongoing debate between those associated with the pet reptile industry, who claim mortality rates are low, and those questioning the acceptability of current practices, who propose that estimates are most likely underestimated due to the mortality rates during trade and deficient record keeping [4,11]. Nevertheless, even the most optimistic scenario evokes the question of whether a 3.6% first-year mortality rate would be perceived as acceptable for cats or dogs. It has been suggested that biased media attention and scientific reporting towards occurrences of abnormal mortality in homes and trade may overestimate the negative aspects of reptile pet keeping [3,9,12,13]. While a reasonable consideration, this claim does not seem to find support in the trade industry that, at least in some cases, refers to mortality rates as high as 70% as industry standard [2,11,14]. 

Research has demonstrated that vertebrates and some invertebrates are sentient, and capable of consciously experiencing positive and negative states, feelings and emotions such as pain, anxiety or pleasure, which ‘*matter to them*’ [15]. A substantial body of research places reptiles among the animals capable of experiencing these subjective states [16,17,18]. Consequently, the ethical debate about reptile keeping needs to take into account the effects on each animal’s health and welfare from birth or capture to death, and how our actions affect their chance of experiencing a ‘*life worth living*’ [19]. Given their duration and prevalence, husbandry problems in pet reptiles deserve at least equal consideration than arguably more visible welfare issues such as mortality during capture or transport.

Keeping exotic pets presents significant challenges when compared to keeping domestic species [5] that have undergone selection for traits that favor their co-existence with humans [20]. Reptiles are ectotherms, meaning that their body temperature is highly dependent of environmental conditions, which affect all physiological and behavioral processes [21,22]. Many reptile species have adapted and evolved to live in environments with highly specific conditions that are difficult to recreate in captivity [23]. This leads to the question of whether even experienced hobbyists and zoos are able to consistently provide lives for these animals that amount to more than just ‘*controlled deprivation*’ [4,18]. Confinement to an enclosure has been referred as a major difference between reptiles and domestic animals in terms of their suitability for captivity [5]. This is especially relevant in ectotherms due to the existence of life-sustaining behavioral mechanisms required to maintain homeostasis, such as behavioral thermoregulation [23] that can be thwarted in insufficiently complex captive environments or enclosures that are too small to provide an appropriate thermal gradient [6].

Knowledge of the behavioral complexity of reptiles is rapidly increasing, with accounts of the expression of play [18] and the ethological need of locomotor [6] and foraging behavior [24]. Increasing evidence has disproven pre-conceived anecdotal (or ‘*folklore*’) knowledge claiming that reptiles are generally sedentary or do not require large or diverse environments [6]. Nevertheless, the knowledge base required to meet the environmental needs of hundreds of reptile species kept in captivity is so extensive (or maybe unavailable) that it remains a challenge for commercial zoos, and is likely beyond the reach of the average pet owner [2,18]. For example, with several hundreds of reptile species observed in the pet trade [4], only a fraction of the species-specific temperature and humidity ranges are available in peer-reviewed literature. Consequently, pet owners often need to rely on the experience of others or on information disseminated by the pet industry to care for their animals. Even veterinarians refer lack of information as a major barrier to the provision of veterinary care to reptiles [7]. As a consequence, if we intend to move beyond reptile pets’ basic needs and towards providing them with the conditions required for a *life worth living*—with full consideration of behavioral repertoire, affective states and emotions—more should be done to bridge the knowledge gaps directly related with captive welfare issues.

Survival-related factors such as nutrition, environment and health influence affective states, and therefore the mental states and welfare of animals [19]. In captive animals, and particularly ectotherms, these factors are heavily dependent on husbandry practices. Therefore, estimates reporting that more than 70% of reptile illnesses are caused by poor husbandry [25] imply poor welfare states associated with nutrition and environment. Metabolic bone disease associated with poor diet, lighting and temperature [26], rostral abrasions associated with interaction with enclosure boundaries, thermal burns, bites from prey and intestinal impactions related to pica or inactivity are some examples of husbandry-derived welfare issues that have been described in captive reptiles [5]. Additionally, knowledge on behavior indicators of reptile welfare [5] and evidence of the behavioral impacts of deficient husbandry [6,27,28] are quickly growing. While providing their pets positive welfare and a ‘*life worth living*’ is currently a mainstream goal for feline and canine companion animals, evidence suggests that many reptile owners might still be struggling to keep their animals pets alive. 

Survey-based studies are increasingly used to investigate exotic pet welfare [29,30,31]. Despite the limitations inherent to self-reporting and convenience sampling, survey data allow an exploratory approach to complex phenomena such as pet reptile welfare, where direct access to animals in private homes is challenging [30]. In this study, we analyze data on behavioral knowledge and husbandry provision reported by reptile owners in an online survey to infer whether the needs of pet reptiles are being met. We also analyze factors influencing owner reports of human-animal bond type and welfare scores. We hypothesized that significant shortcomings would be identified in (1) pet owners’ knowledge of reptile behavior and (2) the provision of essential husbandry needs.

## 2. Materials and Methods

### 2.1. Survey Development and Data Collection

An online survey was designed based on a literature review to gather data on reptile owners’ knowledge of reptile behavior and husbandry practices, following procedures used in previous studies in rabbits [32] and dogs [33]. The first section of the survey included questions describing the respondents’ animal. The second section focused on husbandry practices included questions regarding enclosure barriers, presence and type of light and heat sources, enclosure equipment, temperature and humidity ranges, three-dimensional enrichment, refuges, substrate, enclosure hygiene and diet. The third section included questions on the human-animal bond and owners’ assessment of their pets’ welfare. The fourth section respondents were asked to identify the causes of ten reptile behaviors, and in the fifth section collected owners’ demographic information. The electronic survey was generated using Google Forms and a draft version was piloted by five independent veterinary exotic practitioners, whose comments and suggestions were incorporated into the final revised version.

The survey was submitted via e-mail to owners of pet reptiles attending an exotic veterinary practice (Centro Veterinário de Exóticos, Porto, Portugal) between November and December, 2017. It was also advertised in dedicated reptile forums on social media between November 2017 and February 2018, and remained online until July 2020. Participants were asked to answer the questionnaire only once and having in mind only one animal. Repeated submissions (N = 3) were identified and deleted. For the purpose of the survey, the following definition of welfare was provided: “*the animal is free of hunger, thirst, discomfort, pain, injury, disease and fear, and has the freedom to express its normal behavior*”. The full questionnaire is provided in Appendix A. 

### 2.2. Data Preparation

Data cleaning was performed on survey responses using Microsoft Excel. Ambiguous responses were resolved by author consensus and, when in doubt, the authors erred on the interpretation that suggested the most positive animal welfare. The rationale behind this decision was to avoid the risk of overestimating problems associated with reptile keeping. For example, if a respondent, in a close-ended question, stated there was no heat source in the enclosure, but later in the open response said there was a heat lamp, the answer was considered as having a heat source. Where information was insufficient or author consensus was not possible, the response was grouped into the category “other” for descriptive statistics and excluded from the inferential analysis. A table with details on data cleaning is presented in Appendix A.

The question assessing the human-reptile bond allowed multiple answers among the options “*family-member*”, “*burden*”, “*pet*”, “*friend*” and “*no opinion*”. For descriptive statistics, all answers were reported as proportions relative to the total number of respondents. For further statistical analyses, only the answer reflecting the highest level of attachment was taken into account considering the following order: “*family-member*” > “*friend*” > “*pet*” > “*burden*”). For the assessment of the factors influencing the type of reptile-human bond, bond type was coded as a binomial variable with two categories: “*family-member*” and “*non-family member*” (includes “*burden*”, “*pet*” and “*friend*”). For the assessment of the factors influencing the owner-reported welfare score, welfare answers were coded as a binomial variable with two categories “*good or very good*” and “*average, poor or very poor*”. 

Descriptive statistics were calculated for all questions (Appendix A) and the most illustrative examples of challenges in reptile husbandry are presented in Section 3.2. In order to obtain indicators of the provision of husbandry needs, the most biologically relevant variables for all reptile species were selected by author consensus. The variables “temperature range”, “access to light”, “diet” and “access to refuge” were selected for all species. Each answer was then coded as “adequate” or “inadequate”. Where available, preferred temperature ranges were obtained from peer-reviewed books of reptile medicine and husbandry for 18 of the 49 species named by respondents in this study [34,35,36,37]. For species where references where not available in peer-reviewed literature, references were collected from websites and publications specializing in reptile care. In cases of several different ranges, the lowest and highest temperatures reported for each species were used in order to obtain the most inclusive reference range to minimize the risk of overestimating husbandry deficiencies. Minimum temperatures less than 2 °C below the reference range were considered adequate if there was an adequate high-end temperature thus providing an appropriate gradient. In turn, high-end temperatures more than 2 °C below the reference value were considered inadequate due to risk of chronic hypothermia, and high-end temperatures 5 °C or more above the reference were considered an increased risk for hyperthermia or thermal injury. Snakes were considered as requiring access to light, but not specifically unfiltered UVB light, while lizards and chelonians were considered to require access to unfiltered UVB light (sun light or artificial). Regarding diet, snakes were considered to require whole prey items, while lizards and chelonians were considered to require some form of calcium and vitamin D supplementation (supplements or formulated feed). Access to refuge was considered a need for all reptile species. These criteria are a simplification established by author consensus for the purpose of investigating the specific research question and are likely to underestimate the husbandry needs of most reptiles. Answers were coded as “adequate” or “inadequate” based on whether they reported providing these basic needs. To obtain a single husbandry variable valid for all species, a score was created where 1 point was added for each count of “adequate” 0 points were added for each count of “inadequate” in each of the previously selected husbandry variables, leading to a husbandry score from 0 to 4. Although biologically relevant, the variable humidity was excluded from the inferential analyses due to the differences between aquatic and terrestrial reptiles that rendered a single assessment approach for all species impractical.

In order to obtain indicators of owners’ knowledge of reptile behavior, a set of non-ambiguous questions were selected by author consensus. Selected variables were exploratory behavior, basking behavior, repetitive interaction with enclosure boundaries, anorexia or lethargy, human-directed aggression, open-mouth breathing and defensive cloacal discharge or regurgitation. Each question contained an option where respondents were asked to classify the behavior as “*normal*” according to the following definition: “*a behavior that is natural for the species and related to good animal welfare*”. Owner responses that considered behaviors indicative of poor welfare or negative affective states (repetitive interaction with boundaries, anorexia or lethargy, human-directed aggression, open-mouth breathing and defensive cloacal discharge or regurgitation [5]) to be “*normal*” were coded as “inadequate” interpretations of the behavior. For example, if an owner answered that rapid open-mouth breathing with an extended neck was a normal behavior, the response was classified as an “inadequate” interpretation. The opposite rationale was applied for behaviors associated to good welfare or positive states (exploratory and basking behavior). In order to obtain a single variable reflecting the owners’ knowledge of reptile behavior, a score was created where each “adequate” response added the value of 1 and each “inadequate” response added the value of 0, resulting in a score from 0 to 7.

### 2.3. Statistical Methods

A combination of descriptive analysis of survey answers and inferential analyses using regression models was used to investigate potential gaps in self-reported behavioral knowledge and shortcomings in husbandry provision. Descriptive statistics were calculated using dummy coding and presented for each question, as proportions. This is a common procedure for dealing with survey data from questions where multiple answers are allowed. As several answers were possible for each question, the sum of the results for all the answers of each question may exceed 100%. Inferential analyses were performed using R v. 4.0.3 [38] and significance was set at α = 0.05. Model building followed three steps: (a) purposeful selection of independent variables to include in candidate full models based on their predicted relevance, (b) stepwise variable selection based on likelihood ratio test, and (c) verification of model assumptions. Results are reported as odds ratios, with 95% confidence intervals and *p*-values (OR, 95% CI, *p*-value).

A logistic regression model was built to investigate whether the odds of reporting pet reptiles as family-members (vs. non-family-member) was influenced by reptile group or responder characteristics, namely age, sex, education, environment (urban or rural) and type (breeder or pet owner). A logistic regression model was also used to investigate whether the odds of reporting pet reptile’s welfare as good or very good (compared to average or lower) was influenced by a family-member bond, reptile group, the provision of husbandry needs (husbandry score), knowledge of behavior (behavior score), provision of routine veterinary care, and owner’s education. For logistic regression models, multicollinearity issues were tested for using variance inflation factor (VIF) with a cut-off value of 2, using function car::vif [39]; the response distribution and linearity assumptions were checked by visual inspection of the Q-Q and residuals vs. fitted plots, respectively; outliers were checked for by visual inspection of the residuals vs. leverage plot and Cooke’s distance.

A proportional odds regression model was used to assess whether the knowledge of reptile behavior (behavior score) was influenced by bond type, taxonomic group, routine veterinary care, reptile source or owner characteristics. Finally, a proportional odds regression model was used to investigate if the provision of husbandry needs was affected by bond type, knowledge of behavior (behavior score), taxonomic group, provision of veterinary care, reptile source and owner characteristics. For proportional odds models, the proportional odds assumption was verified using the function brant::brant [40]; goodness of fit was tested with an ordinal version of the Hosmer–Lemeshow test, the Lipsitz test and the Pulkstenis–Robinson [41], using the R. package “generalhoslem” [42].

## 3. Results

### 3.1. Sample Characteristics

A total of 220 respondents from Portugal keeping reptiles as companion animals answered the survey (Table 1). Among them, 65% kept chelonians, 19% kept lizards and 16% kept snakes. Most identified themselves as pet owners (95%), while 5% described themselves as breeders. Fifty-six percent of respondents were female (1% did not report gender). The age of the respondents ranged from 16 to 65, with a median of 27 and an inter-quartile range of 15. Eighty-three percent of the respondents lived in an urban environment, whereas 15% lived in a rural environment (2% did not answer). Fifty-eight percent of respondents had a higher education (Degree, MSc. or PhD.), 35% had completed high-school and 6% had completed the 9th grade or less. Reptiles were mostly sourced from pet shops (65%), received as gifts (23%) or acquired from a breeder (10%). A small number was adopted/rescued (1%) or collected from nature (1%). Routine veterinary care was provided to 42% of reptiles, 53% received veterinary care only in case of illness, and 5% never visited the vet. Companion reptiles were most frequently considered family-members (64%), pets (28%) or friends (8%). A single respondent considered the reptile a burden.

### 3.2. Reptile Husbandry

A direct source of UVB light or sunlight was reported absent for 56% of the lizards and 29% of the chelonians (Figure 1). In snakes, 77% of the owners reported no direct UVB light source, and 17% reported no light source at all. 

Transparent enclosure barriers were the most common across all taxonomic groups. Tri-dimensional enrichment (specifically “*branches and furniture that allow climbing*”) was reported for 54% of the snakes and 78% of the lizards. Sixteen percent of the chelonians were terrestrial, and the remaining were semi-aquatic, most of which (97%) had access to both water and dry land. 

The provision of refuges in the enclosure was reported for 91% of the snakes, 85% of the lizards and 69% of the chelonians. The choice of substrate varied greatly, and enclosures were most frequently cleaned one to six times per week for all reptile groups. 

Reported diets were also varied, with live prey used infrequently (11% of snakes, 17% of lizards). Dietary supplementation was not ubiquitous, with 88% of the lizards and 35% of the chelonians reported as not receiving vitamin D or calcium supplements in any form. 

Reported temperature ranges were consistent with available information and therefore classified as adequate for 69% of the snakes, 41% of the lizards and 19% of the chelonians (Figure 1). Bar plots displaying frequencies for all husbandry questions are available in the Appendix A.

### 3.3. Knowledge of Reptile Behavior

Approximately 90% of all respondents considered exploratory behavior a normal behavior; 14% of snake keepers, 29% of lizard keepers and 31% of chelonian keepers also considered it an attempt to communicate. Approximately 80% of respondents considered basking behavior normal. Some owners (39% lizards and chelonians, 49% snakes) also related basking behavior to thermoregulation. Repeatedly hitting the head against—or trying to climb—the enclosure glass or wall was considered normal by approximately one third of the keepers of all reptile groups (31% snakes, 29% lizards, 27% chelonians). Lethargy and reduced intake were associated with many causes, but most frequently to hibernation or pain, discomfort or illness. Aggression toward humans was considered normal by more than one quarter of reptile owners, and most frequently motivated by stress or fear (80% snakes, 78% lizards, 58% chelonians). Rapid open-mouth breathing similar to panting, with an extended neck was considered normal by a minority of respondents (17% snakes, 12% lizards, 8% chelonians) and was most commonly associated with discomfort or illness. Cloacal discharge or regurgitation in response to human contact or presence was considered normal by 23% of the snake keepers, 15% of the lizard keepers and 11% of the chelonian keepers. The behavior was associated with stress or fear in more than half of the cases. Moving to a dark area of the enclosure or refuge was considered normal by 91% snake keepers, 71% of the lizard keepers and 53% of the chelonian keepers; it was most frequently associated with stress or fear in snakes and chelonians, and to thermoregulation in lizards. In snakes, difficulty to coil was considered normal by 14% of the respondents, and associated to pain, discomfort or illness (71%). In lizards, puffing-up was considered normal for 24% of the keepers and mostly associated with stress or fear (63%) or attempts to communicate (24%). Retracting into shell was considered normal by 34% of chelonian keepers and was most frequently associated with stress or fear (83%). Results are summarized in Figure 2. Bar plots displaying frequencies for all behavior questions are available in the Appendix A.

### 3.4. Factors Associated with Reporting a Family-Member Bond

The variables that significantly improved model fit were education level and taxonomic group (Table 2). People with higher education had 51% (OR = 0.49, 95% CI = 0.26–0.89, *p* = 0.02) lower odds of reporting a family member bond. In terms of taxonomic group, the odds of lizard keepers considering their reptile a family-member were 2.40 times higher (OR = 2.40, 95% CI = 1.04–5.58, *p* = 0.04) than chelonian keepers, while accounting for gender and education level (Table 3). The odds of snake keepers considering their reptiles family members did not differ significantly from chelonians. Gender did not significantly improve model fit, but was maintained to control for its influence on human-animal relations. In the final model, there was some evidence supporting an effect of gender, where male respondents had 46% (OR = 0.54, 95% CI = 0.30–1.00, *p* = 0.05) lower odds of reporting a family member bond.

### 3.5. Factors Associated with Reporting Good/Very Good Welfare

None of the respondents reported their reptile’s welfare as very poor, and a single respondent reported it as poor. The majority of the reptile owners considered their pets’ welfare to be good (43%) or very good (43%), while 14% of them reported it to be average (Table 1). Husbandry score, bond type and taxonomic group significantly improved model fit when exploring the variables influencing the odds of reporting good/very good welfare (Table 2). For each unit increase in husbandry score, the odds of reporting good/very good welfare were 2.11 times higher (OR = 2.11, 95% CI = 1.26–3.54, *p* < 0.01). Owners reporting a non-family-member bond type had 61% lower odds (OR = 0.39, 95% CI = 0.17–0.90, *p* = 0.03) of reporting welfare as good/very good (Table 3). Lizard owners had 8.06 times higher odds (OR = 8.06, 95% CI = 1.04–62.23, *p* = 0.04) of reporting welfare as good/very good compared to chelonian owners. The odds of snake owners reporting good/very good welfare did not differ significantly from chelonian owners.

### 3.6. Factors That Influence the Interpretation of Reptile Behavior

There was strong evidence for an effect of respondent age and environment on behavioral scores (Table 4). In the final model, for each one year increase in age (ranging from 16 to 65), the odds of scoring a one point higher on the behavioral score were 4% lower (OR = 0.96, 95% CI = 0.94–0.99, *p* < 0.01). For respondents living in urban areas, the odds of scoring one point higher on the behavioral score were 53.3% lower (OR = 0.47, 95% CI = 0.22–0.97, *p* = 0.04) compared to respondents from rural environments (Table 5). Gender did not significantly improve the model but was retained to control for its potential influence on how reptile behaviors were perceived. Male respondents had 42% lower odds of scoring one point higher on the behavioral score (OR = 0.57, 95% CI = 0.34–0.94, *p* = 0.03) compared to females. Education level, taxonomic group, reptile source, provision of routine veterinary care and a family-member bond did not significantly improve the model explaining behavioral scores.

### 3.7. Factors That Influence the Reported Provision of Husbandry Needs

There was strong evidence for an effect of reptile taxonomic group and pet owner age and gender on husbandry scores (Table 4). For snake keepers, the odds of scoring a one point higher on the husbandry score was 14 times higher than chelonian keepers (OR = 14.05, 95% CI = 5.94–34.96, *p* < 0.01). For lizard keepers, the odds of scoring one point higher on the husbandry score did not differ from chelonian keepers (OR = 1.34, 95% CI = 0.66–2.73, *p* = 0.42). For age, there was a 6% increase of the odds of scoring one point higher in husbandry provision with each additional year (OR = 1.06, 95% CI = 1.03–1.09, *p* < 0.01). Male respondents had 1.89 times higher (OR = 1.89, 95% CI = 1.08–3.33, *p* = 0.03) odds of scoring one point higher on the husbandry score, compared to female respondents (Table 5). The behavioral score did not significantly improve the model explaining husbandry score, but was kept in the model due to the importance that understanding behavioral signs associated with poor welfare has in identifying and correcting husbandry. A trend was observed suggesting a 20% increase in odds to score one point higher in husbandry provision, for each additional point scored in behavioral questions (OR = 1.20, 95% CI = 0.98–1.47, *p* = 0.08). Respondents’ education level and environment, reptile source, provision of routine veterinary care and a family-member bond did not significantly improve the model explaining husbandry scores.

## 4. Discussion

In this study we used survey data to assess behavioral knowledge and husbandry provision reported by pet reptile owners in Portugal. We predicted that owners would be largely unskillful at interpreting normal and abnormal reptile behaviors and that husbandry needs essential to the survival of pet reptiles would not be met. The first of these predictions was not confirmed: Two-thirds of respondents (68%) had behavioral scores of 6 or 7, indicating very good to excellent knowledge of their pet reptile’s behaviors, and only 15% scored 4 or lower. Conversely, only 15% of respondents reported providing all of the four survival-related husbandry needs included in our analysis (temperature, lighting, diet and refuge) and 43% met two or less (Table 1). These findings support the prediction that many pet reptiles are not being provided with some of the basic survival-related needs, and are likely to experience unrecognized poor welfare for a significant proportion of their life. Notwithstanding methodological limitations, this should be seen a best-case scenario, since the population of our inquiry is arguably composed of educated, informed, and motivated owners that were willing to share their knowledge and opinions on pet reptile keeping and to seek veterinary care for their animals.

Overall, 85% of the survey participants (Table 1) reported failing to provide at least one of the four basic requirements used to calculate the husbandry score (lighting, temperature, diet and refuge). Failure to provide a direct source of UVB light and an adequate temperature gradient for the species in over 30% of the cases are two crucial examples that directly impact reptile health and survival. Baines et al. eloquently draw attention to the dependence of reptiles on these factors by stating that ‘*Every aspect of the life of a reptile or amphibian is governed by its daily experience of solar light and heat*’ ([43], p. 42). In the case of lizards and chelonians, UV light is necessary for the behaviorally regulated production of vitamin D [44] and its absence is causally linked to metabolic bone disease, which limits the quality of life of reptiles by affecting their ability to move and express normal behavior, and may cause death [26]. This risk is further increased in the absence of dietary supplementation, which was reported by 88% of the lizard owners. Inadequate temperature gradients inhibit behavioral thermoregulation [45] that is necessary for fundamental biological processes such as digestion and immunity [46,47]. Failure to provide these environmental needs is also very likely to result in behavioral deprivation and negative affective states. Exposure to UV light has been shown to induce β-endorphin secretion in human skin that is associated with feelings of pleasure and reward to the extent of addiction [48]. It is possible that basking in UV light is associated with feelings of thermal comfort and endorphin-mediated feelings of pleasure in reptiles. Indeed, even snakes that are not traditionally thought to require a UVB light source have been recently shown to require a UVB light source to express circadian behavioral cycles and basking behavior [27]. According to our results, most snakes and lizards had access to refuges, but almost a third of the chelonians did not. This difference could be due to the high representation of freshwater turtles (e.g., *Trachemys* spp., *Graptemys* spp.) that are typically sold at a low cost with small aqua-terrarium setups, while snakes and lizards are usually acquired at a higher cost and with more complex setups. We were unable to confirm this suspicion because minimal data was collected in relation to enclosure complexity, and the adequacy of the enclosure is highly dependent on data not included within the survey (e.g., specimen size and concurrent husbandry practices). 

Despite indication that many reptiles are not being provided with the most basic survival-related husbandry needs, 86% of respondents reported that their pets experienced good or very good welfare. Self-reported welfare and husbandry scores were also conflicting when looking at the different reptile groups. While snake owners had fourteen times higher odds of reporting adequate husbandry provision, lizard owners had the highest odds of reporting good or very good welfare despite providing less of their animals’ basic husbandry needs. Husbandry scores were also significantly influenced by reptile group, owner age and gender. Access to routine veterinary care, a family member bond and behavioral knowledge did not significantly impact the self-reported provision of basic husbandry needs. Veterinarians have acknowledged that the lack of information on the care of many exotic pets is so profound that even they are unable to provide owners with advice for some species [7]. 

Self-reported behavioral knowledge scores were negatively influenced by respondent age and living in an urban environment. Kellert [49] found that children from rural areas were more interested and knowledgeable about animals compared to those from urban settings, which is consistent with this finding. However, perhaps due to selection *bias* inherent to the online data collection method [50,51], our sample is heavily skewed toward young urban respondents, which limits further interpretations. The generally high behavioral knowledge score and absence of its effect on husbandry supports the suggestion that behavioral knowledge is not a limiting factor for reptile welfare in this study population.

Several behaviors that have been associated with poor welfare and captivity stress were considered ‘*normal: natural for the species and related with good animal welfare*’ by up to one quarter of survey respondents (Figure 2). Notable examples were repeatedly hitting the head against—or trying to climb—the enclosure glass or wall, aggression toward humans, rapid open-mouth breathing with an extended neck, and cloacal discharge or regurgitation in response to human contact or presence. Aggression toward humans, cloacal discharge and regurgitation are all behaviors associated with stress or fear [5] and were recognized as such by two thirds of the respondents. Interaction with enclosure boundaries is related with captivity-stress, and can occupy large amounts of active time and lead to rostral lesions [5]. In a study comparing ball python (*Python regius*) behavior between small barren and large enriched enclosures, only snakes in barren environments exhibited this type of behavior, which was accompanied by reduced behavioral diversity and resolved when moved to an enriched enclosure [27]. A large proportion (above 90%) of the respondents in this study believed this behavior to be caused by stress, fear or an attempt to escape, which apparently contradicts the answers defining it as ‘*normal*’. Similarly, rapid open-mouth breathing, which is associated with hyperthermia [5], was also frequently considered ‘*normal*’. 

Altogether, these findings suggest that behaviors associated with negative welfare in pet reptiles are adequately interpreted by most owners, but nonetheless considered ‘*normal*’ by some. One possible explanation is that the frequency of these behaviors in pet reptiles has led to their acceptance as the norm, and hampered the search for ways to prevent them. The normality of these (otherwise abnormal) behaviors may lead to the desensitization to animal suffering which, in turn, may lead to failure to adequately care for these animals. A similar concern has been reported for dogs of brachycephalic breeds, whose owners fail to recognize clinical signs of brachycephalic obstructive airway syndrome as a welfare problem [52]. Due to the frequency of the signs among animals of those breeds and their presentation since birth, owners seem to perceive them as normal for the breed or cute. Likewise, reptile owners in our study defined their animals as family-members and rated their welfare good or very good, but failed to provide them with some essential husbandry needs. Despite evidence indicating that this cohort of respondents has strong emotional bonds towards their animals [53] the owners’ acceptance of substandard care (or normality) limits their need to search for improved procedures that promote reptile welfare. This finding suggests that reptile owners believe that their animals fare well despite simultaneously failing to provide them with all of their basic husbandry needs or failing to recognize displayed behaviors relating to negative welfare. Possible explanations could include denial, mental heuristics, ignorance or practical constraints.

While it might be an unrealistic expectation to fully replicate natural environments, our results indicate that many pet reptiles in Portugal live in, at best, controlled deprivation. The concept of ‘*controlled deprivation*’ was coined by ethologist Gordon Burghardt in the 1990’s to recognize that “*all captive environments deprive animals of some natural stimuli and that these restrictions have varying, and often unpredictable, consequences on the welfare of captive animals*.” ([54], p. 264). This unpredictability is magnified in reptiles because, contrary to the advertised messaging by the pet trade industry and by common assumptions portraying some species as “*easy to keep*” or “*adequate for beginners*”, all reptiles require complex species-specific husbandry conditions, including temperature gradients, UVB light, dietary needs and environmental enrichment [3,17,42]. Animals that are prevented from performing species-specific behaviors, that exhibit stereotypic or aggressive behaviors, or that are likely to be trying to escape their enclosure, experience negative mental states [55].

Consumer behavior needs to be taken into account when promoting responsible reptile pet ownership. In particular, the predominant view, reinforced by the exotic pet industry, that reptiles are low-maintenance pets needs to be actively refuted. However, Moorhouse et al. [56] found that animal welfare concerns and conservation impacts were unlikely to significantly influence consumers of exotic pets. In turn, providing consumers with pre-purchase information regarding zoonotic potential and legal consequences decreased the likelihood of purchasing an exotic pet in 39% [56]. Those findings suggest there is efficacy in campaigns that adopt an anthropocentric view, focusing on the interests of the owner, how keeping a reptile is difficult, laborious and time consuming, and how improved welfare can reduce zoonotic risk. On the other hand, our results show that owners are capable of recognizing behaviors associated with poor welfare but seem to accept them as normal, which highlights a potential strategy for campaigns to use a zoocentric approach which challenges the desensitization toward poor reptile welfare.

The challenges of identifying animal-based methods to directly assess the welfare of captive reptiles have been acknowledged by a recent review [57]. Due to the challenges of accessing reptiles in private homes, self-reported welfare scores are useful to provide some insight into pet reptile welfare. Self-reported welfare scores in our study were also influenced by a family member bond and by reptile group, which suggests that future studies using self-reported welfare assessments in reptiles should account for these potentially confounding effects. The type of bond with the reptile influenced owner welfare scores, but not behavioral and husbandry scores, which could signal a lower reliability of self-reported welfare ratings compared to survey questions targeting objective environmental and animal-based parameters. This discrepancy could be driven by standards of welfare that are biased by folklore knowledge [58] or social desirability *bias* [59] and hamper accurate welfare assessments. Further studies are required to clarify these relations.

In terms of factors associated with human-animal bond, it is somewhat surprising that higher education decreased the odds of reporting a family member bond. A possible explanation is that respondents with higher education might better understand that reptiles, as non-domesticated wild animals, may actually benefit from a more distanced relation with humans. An alternative explanation is that this variable could be confounded by factors that were not captured by the questionnaire, such as loneliness [60,61] or mental health [62]. There was also a small gender effect in which men were slightly less likely to consider their pet reptile a family member. This trend has also been reported for dogs and cats [63].

The methodological limitations of this study need to be considered when interpreting these results. For example, it is possible that some respondents may have confused the given definition of ‘*normal*’ with other connotations such as ‘*frequent*’ as they progress through the questionnaire. However, the high prevalence of husbandry-related illness in clinical practice [3,34] and the high mortality during the first years in captivity [9] are consistent with our findings. Other limitations include the low sample size for lizards and snakes and the lack of an indicator of behavior frequency, which is in some cases relevant to determine whether the behavior is associated with good or poor welfare. Further studies relying on direct observation and including physiological indicators of reptile welfare would be useful in clarifying these findings. 

## 5. Conclusions

This study presents an indirect assessment of the welfare of reptiles kept as companion animals in Portuguese homes. Self-reported data showed that most reptile owners in Portugal fail to provide their pets with at least one survival-related husbandry need, while rating their welfare as good or very good. Despite correctly recognizing and interpreting behavioral signs of poor welfare, owners often consider them normal for the species. These results suggest that many reptiles in Portugal could be suffering unrecognized poor welfare throughout their lives as pets. This is in conflict with the reported well-intended motivations of most reptile owners participating in this survey [53]. Desensitization toward animal suffering driven by misconceptions perpetuated by the pet trade industry could be involved. Overall, these findings emphasize the importance of focusing awareness campaigns on resetting the norm for what is acceptable in terms of reptile welfare.

## Figures and Tables

**Figure 1 animals-11-02964-f001:**
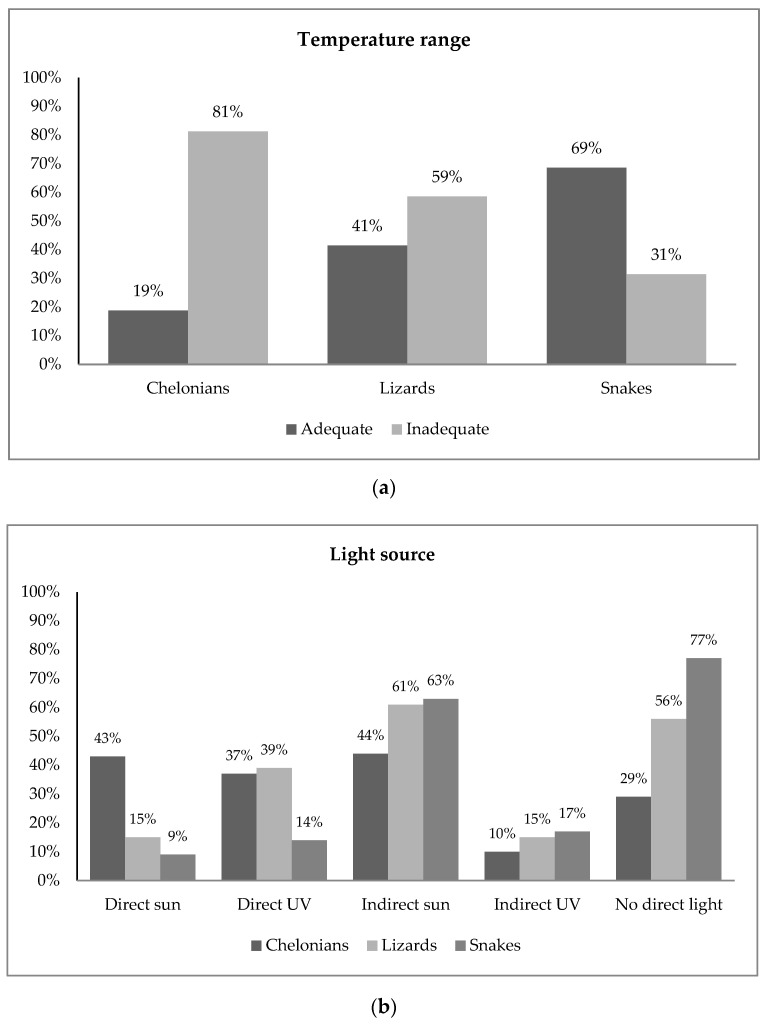
Frequency plots of owner-reported data for (**a**) temperature range and (**b**) light source in snakes, lizards and chelonians. The data indicate limited access of pet reptiles to environmental conditions directly related with survival. Bar plots displaying frequencies for all husbandry questions are available in the Appendix A (Appendix A).

**Figure 2 animals-11-02964-f002:**
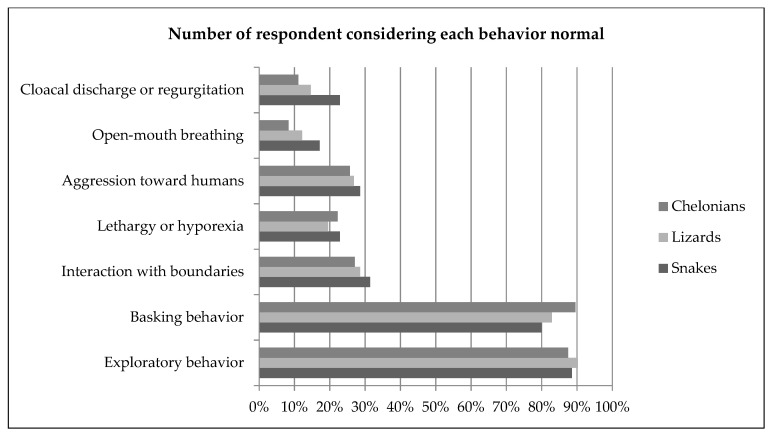
Proportion of owners considering selected behaviors normal (i.e., “*normal for the species and associated with positive welfare*”) in snakes, lizards and chelonians. Bar plots displaying frequencies for all behavior questions are available in the Appendix A (Appendix A).

**Table 1 animals-11-02964-t001:** Frequency tables for categorical (A) and ordinal (B) variables, and median and min-max range for continuous variable age (C).

**A. Categorical Variables**	**Level**	**Count**	**Frequency**
	Chelonians	144	0.65
Group	Snakes	41	0.19
	Lizards	35	0.16
	Breeder	23	0.10
	Gift	50	0.23
Source	Nature	2	0.01
	Pet shop	142	0.65
	Rescued	3	0.01
	Burden	1	0.00
Human-animal	Pet	62	0.28
Bond	Friend	17	0.08
	Family-member	140	0.64
Owner-reported welfare	Very poor	0	0.00
	Poor	1	0.00
	Average	25	0.14
	Good	91	0.43
	Very good	89	0.43
	Breeder	11	0.05
Owner type	Pet owner	208	0.95
	NA	1	0.00
	Male	95	0.43
Gender	Female	123	0.56
	NA	2	0.01
	Rural	33	0.15
Environment	Urban	183	0.83
	NA	4	0.02
	<9th	3	0.01
	9th	11	0.05
	12th	78	0.35
Education	Degree	85	0.39
	MSc.	39	0.18
	PhD.	3	0.01
	NA	1	0.00
	Never	11	0.05
Veterinary care	Illness	116	0.53
	Routine	93	0.42
**B. Ordinal Variables**	**Level**	**Count**	**Frequency**
Behavioral score	2	9	0.04
	3	11	0.05
	4	14	0.06
	5	35	0.16
	6	73	0.33
	7	78	0.35
Husbandry score	0	3	0.01
	1	13	0.06
	2	78	0.35
	3	92	0.42
	4	34	0.15
**C. Continuous Variables**	**Median**	**Range**	**No answer**
Age	27	16–65	12

**Table 2 animals-11-02964-t002:** Variable selection using single term deletions from the global models for reporting a family member bond and a good or very good welfare score (* denotes statistically significant results).

Dependent Variable	Factor	Df	Deviance	AIC	LRT	*p*-Value
Bond type	Group	2	255.66	265.66	7.35	0.03 *
	Age	1	250.23	262.23	1.92	0.17
	Gender	1	251.05	263.05	2.73	0.10
	Environment	1	249.27	261.27	0.95	0.33
	Education	1	253.23	265.23	4.92	0.03 *
Welfare score	Bond	1	149.64	163.64	4.79	0.03 *
	Group	2	152.64	164.64	7.79	0.02 *
	Husbandry score	1	151.44	165.44	6.59	0.01 *
	Behavior score	1	144.87	158.87	0.03	0.87
	Veterinary care	1	146.02	160.02	1.17	0.28
	Education	1	146.84	160.84	1.99	0.16

**Table 3 animals-11-02964-t003:** Estimated coefficients, odds ratios and 95% confidence intervals of the final logistic regression models for reporting a family member bond and a good or very good welfare score (* denotes statistically significant results).

Dependent Variable	Factor	Estimate	Std. Error	z-Value	*p*-Value	Odds Ratio	2.5%	97.5%
Bond type	Group (lizards)	0.88	0.43	2.04	0.04 *	2.40	1.04	5.58
	Group (snakes)	−0.56	0.41	−1.36	0.17	0.57	0.26	1.28
	Gender (male)	−0.61	0.31	−1.96	0.05 *	0.55	0.30	1.00
	Education (higher)	−0.72	0.31	−2.32	0.02 *	0.49	0.27	0.90
Welfare score	Bond (non-family)	−0.93	0.42	−2.20	0.03 *	0.39	0.17	0.90
	Group (lizards)	2.09	1.04	2.00	0.05 *	8.06	1.04	62.23
	Group (snakes)	1.66	1.08	1.54	0.12	5.26	0.64	43.30
	Husbandry score	0.75	0.26	2.84	4.54 × 10^−3^ *	2.11	1.26	3.54

**Table 4 animals-11-02964-t004:** Variable selection using single term deletions from the global ordinal regression models for behavior score and husbandry score (* denotes statistically significant results).

Dependent Variable	Factor	Df	AIC	LRT	*p*-Value
Behavior score	Bond	1	621.91	0.03	0.87
	Group	2	620.35	0.46	0.79
	Vet care	1	621.97	0.08	0.78
	Age	1	630.53	8.64	3.29 × 10^−3^ *
	Gender	1	625.29	3.40	0.07
	Education	1	622.71	0.82	0.36
	Environment	1	627.18	5.29	0.02*
	Reptile source	2	621.12	1.24	0.54
Husbandry score	Bond	1	471.04	0.81	0.37
	Group	2	498.44	30.21	3.00 × 10^−7^ *
	Vet care	1	470.23	1.23 × 10^−3^	0.97
	Behavior score	1	472.96	2.73	0.10
	Age	1	483.59	13.35	2.58 × 10^−4^ *
	Gender	1	474.08	3.84	0.05 *
	Education	1	470.74	0.50	0.48
	Environment	1	471.81	1.57	0.21
	Reptile source	2	471.48	3.25	0.20

**Table 5 animals-11-02964-t005:** Estimated coefficients, odds ratios and 95% confidence intervals of the final ordinal regression models for behavior score and husbandry score (* denotes statistically significant results).

Dependent Variable	Variable	Estimate	Std. Error	z-Value	*p*-Value	OR	2.5%	97.5%
Behavior score	Age	−0.04	0.01	−2.77	0.01 *	0.96	0.94	0.99
	Environment (urban)	−0.76	0.38	−2.02	0.04 *	0.47	0.22	0.97
	Gender (male)	−0.57	0.26	−2.17	0.03 *	0.57	0.34	0.95
Husbandry score	Group (lizards)	0.29	0.36	0.81	0.42	1.34	0.66	2.73
	Group (snakes)	2.64	0.45	5.86	4.57 × 10^−9^ *	14.06	5.94	34.96
	Age	0.05	0.01	3.73	1.95 × 10^−4^ *	1.06	1.03	1.09
	Gender (male)	0.64	0.29	2.22	0.03 *	1.89	1.08	3.33
	Behavior score	0.18	0.10	1.76	0.08	1.20	0.98	1.48

## Data Availability

The data presented in this study are available on request from the corresponding author. The data are not publicly available due to confidentiality issues.

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
