# Peer review of "Pet Reptiles—Are We Meeting Their Needs?"

_animals, 2021, doi:10.3390/ani11102964_

Round 1

Reviewer 1 Report

The Manuscript animals-1396849 "Pet reptiles – are we meeting their needs?" describes the results of a statistical analysis of an electronic questionnaire for reptile owners and on reptile keeping and behavior. The questionnaire and the statistical analysis are described in detail. All parts of the manuscript are well written and the references are sufficient. The fact that routine veterinary care does not have a significant impact on basic animal husbandry needs is very sad for the veterinary community. It may be useful to mention in the discussion that veterinarians can also specialize in reptile medicine under the umbrella of the EBVS in the form of the ECZM (Herp), which may lead to a significant improvement in advice. However, congratulation to this survey and manuscript, no other requests or comments. 

Author Response

Dear reviewer 1,

We thank you for your kind comments. When considering these results, we need to carefully account for their limitations. The fact that reporting routine care does not impact reported husbandry conditions does not question the quality of veterinary care, despite the limitations we face as vets in providing care for reptiles (e.g. pharmacological information, pathophysiology and even husbandry). It might perhaps say a bit about how much the herpetologist trusts the advice of the veterinarian compared to his peer herpetologist. This relates to the issue of reliance on anecdotal knowledge that has been the information source for decades. No doubt, we need to work to change this paradigm to promote evidence-based decisions, in this area, and many others. Providing specialized professionals backed by the credibility of the EBVS and ECZM will certainly contribute to this goal.

Reviewer 2 Report

This study represents a timely and well-needed assessment of animal welfare among pet reptiles. The survey was well designed and the statistical analysis thorough. The study reveals multiple problems associated with animal husbandry of pet reptiles, and is a most welcome contribution to the growing body of literature that de-objectifies non-primate animals, and reveals systemic shortcomings in husbandry conditions. I hope to see this manuscript in print following minor revisions to its core questions and associated answers.

The central problem of this manuscript is a lack of clarity concerning the terms “welfare requirements”, “essential husbandry needs”, and “a life worth living”. These terms are frequently employed to produce key arguments, and to quantitatively assess the knowledge of pet owners, and the physical constitution of their pets. However, as far as I can tell none of these terms is properly defined, making it difficult to assess the validity of hypotheses and their associated tests. These terms need to be clearly defined, and the associated hypotheses put into a testable context. Outlining key questions, and testing them through evaluation of the four (?) key aspects of husbandry will relieve this manuscript of its current ambiguities in this regard.

Below are a few minor comments regarding wording and clarity.

88: Use “falsified” instead of “demystified”, to increase clarity of writing. Unless that is not what you mean, in which case you need to clarify your statement.

91: Add a qualifying remark and a citation. If your “bar” for keeping reptiles is pampering them to complete satisfaction, then say so. You claim that the knowledge required to keep reptiles is “enormous”, i.e. markedly greater than for common pets such as dogs. Either qualify this statement with a citation, or change it.

E.g.: what is the optimal ambient temperature and humidity for a border collie aged three years? And is this a realistic and well-targeted question, or is this targeting a quality of life that is beyond reasonable reach?

Few terrestrial animals require or even tolerate a constant maintenance of “optimal” environmental conditions. You need to be careful with your wording in this regard, and specific with your definition of “welfare requirements”.

107: Illness is necessarily associated with “illness”. Strike the second mention of the word.

115: As in line 91, you need to define “happiness” or a “life worth living”. Your target cannot be achieved, if you do not define it.

119-126: I prefer the use of Simple Present in the introduction, and Past Tense for conclusion and abstract. Not a must, but it reads a bit cleaner.

130: May we know who those precursors of your survey were designed by, and how they were assessed?

135: You add a completely new line of predecessors to your survey. Now it becomes really important to know which surveys were mentioned in line 129/130.

163: Strike the second word “FIGURE”.

171: It is quite possible that optimal husbandry conditions are reported incorrectly on websites and in other popular literature sources. This seems a fruitful line of future investigations.

273: […] “in” any form?

286: These figures imply a scary degree of ignorance among pet keepers.

355: You seem to be mixing data from pet and pet owner in this sentence, which is quite confusing. Make specific reference to owner or pet when you mix both in one sentence. Also, mind that you report sex for reptiles, and gender for humans. Be consistent to avoid ambiguities.

380: You need to clearly define these “essential husbandry needs”. Do you mean the four categories that you scored with one point each? If so, clearly define this in the introduction, writing “these are the four key factors we define for meeting husbandry needs.

383: Are the “welfare requirements” a subset of the “essential husbandry needs”? This is not clear.

437: strike the “it” at the end of the line.

440: relate this statement to the corresponding bar plot, i.e. reference the figure.

444: how great is the correlation between a) people who misjudge these stress responses with b) enclosures that lack refugia? If you can link these two groups, you can argue that a small knowledge base directly impacts animal welfare.

490: It might be better to dress this political statement in more neutral language.

492: This sentence is rather long, lacks an object, and circles back on itself, making it difficult to read. Try something like: “[…] and suggest that campaigns that aim to increase animal welfare might be more effective when aimed at challenging […]”.

506: Are you suggesting that people with higher education are less lonely? This argument is difficult to follow.

521: One lucrative alternative to the digital questionnaire would be a short check list that the veterinarian completes with every visit. You could record responses to the four  essential husbandry needs from every owner of a pet reptile who visits the veterinary clinic, and, under proper consideration of privacy, correlate these data with animal welfare and/or personal information of the owner.

Reviewer 3 Report

Overall, this is an important topic. Reptile welfare needs further attention, and this data is an important contribution. This article should be published. But I do think it needs some modifications. I have provided extensive information, intended in a positive spirit. Please don’t be disheartened when you see a lot of suggestions…many are easily addressed, some will take a little bit more work to address.

I have provided specific comments below, but I think overall the paper could be written with a tighter structure and the authors need to be careful to be really specific about their conclusions.

This paper will no doubt elicit some pushback from those who wish to keep reptiles, so its important that it is as tight as possible. At times I felt that the conclusions drawn were overstated. For example, it is stated that the study confirms that reptiles are kept under “controlled deprivation”. To counter this, someone only needs to show one reptile kept in excellent conditions. So I would say “many pet reptiles in Portugal”. I don’t think that weakens your point – I think it strengthens your position because that is much harder to counter.

At times I also felt that the language was unclear and I think it some further proofing would improve this paper.

I recommend that the authors read this paper, as I feel that some points are made in it that will strengthen the current discussion: https://www.mdpi.com/2076-2615/9/1/27

One angle that wasn’t explored which I feel would strengthen your argument, and perhaps aid in future campaigns, was the potential for zoonoses being exacerbated by poor reptile husbandry. E.g. inappropriate thermal conditions à immunosuppression à illness à transmission to people.

I also thought further discussion could be had around the recent papers about snakes not being able to extend their bodies in enclosures.

Specific feedback:

Line 17: clarify as outlined above.

Line 20-21: change to “normal, precluding…”

Line 26: reptiles and other animals

Line 26: is it that little is known about the knowledge of reptile owners, or is little known about their ability to meet their animal’s welfare requirements?

Line 32: add “only” a single respondent

Line 38: the word normalcy isn’t really appropriate here. Do you mean the frequency of these behaviours has lead to their acceptance as the norm? The fact that they are common?

Line 47: some references would be great here.

Line 49: I am uncomfortable with the word “most” here when only two references are cited. Several authors?

Line 51: barriers instead of issues?

Line 56: This is a HUGE range and there is a significant documented debate by the authors of each of these respective studies critical of methodology, COIs etc. I don’t think you can simply print this range then say “These numbers are most likely underestimated” – do you mean the 3.6% is an underestimate, or both this and the 75%. I think there is more scope to comment on this range because its part of the issue. The herpetologists claim a low mortality rate (is 3.6% low? Those working in companion animal practice with dogs and cats might think not!) while the anti-reptile keeping researchers report the high figure.

Line 61: which mortality rates?

Line 67: what do you mean by “and ultimately, life”? – doesn’t quite make sense – has the sentence been cut off here?

Line 70: delete “Under this perspective”, and start sentence with “Given…”

Line 78: place the reference at the end of the sentence?

Line 79-80: suggest “consistently provide lives for these animals that amount to more than just …”

Line 85: or large enough to provide an appropriate thermal gradient

Line 91: the phrase “potentially unattainable” doesn’t sit well here. I feel there is a better way to express this point. Is it the knowledge base that is unattainable, or just the ability to meet the requirements (ie action rather than just knowledge)?

Lines 110-115: this is a bit confusing, as you tell us that the knowledge base is growing re reptiles, then move onto the shift to positive welfare in dogs and cats, then contrast this to reptiles. I think this should be two different paragraphs.

Line 116: delete “being”

Line 117-119: I don’t understand how direct observations are not feasible for pet reptiles. They are captive and therefore easily observed. Often more so because they may be kept in barren environments with no place to hide.

Lines 122-124:I think reword this a little to reflect your hypothesis eg we hypothesised that 1) xxxxx and 2 xxxx

Lines 123-126: This needs to be in your methodology section.

Section 2.1 Data collection: it would be good to include a section on survey development before talking about the data collection

Line 146-147: This wasn’t so clear to me. I think what you are saying is that where the response was ambiguous, the authors erred on the interpretation which suggested the most positive animal welfare?

Line 157: burden, pet and friend are grouped together but are these all reflective of not being a family member? This sentence just needs to be clarified.

Line 160-163: this is about the development of the questionnaire. The structure of the methods is a bit messy, try to use a chronological framework, so start with questionnaire development, then structure, then distribution, then data cleaning and analysis…

Line 169-170: were these named by respondents? Or was the study limited to these species?

Line 176: remove healthy replace with “appropriate thermal gradient”

Line 180-181: suggest reword, what do you mean by “at least access to some form of light”? any light? Sunlight? Artificial? For one second or for longer?

Statistical methods: I am not a statistician, so cannot comment specifically on the methods, but the description wasn’t altogether clear for me, eg why was dummy coding needed to calculate descriptive statistics?

Line 242: was this a software package?

For all results, provide the % but also the n=

Line 247: instead of “were”, suggest “identified themselves as pet owner”

Line 265: define “tridimensional enrichment”

Lines 262-277: suggest break this paragraph up into smaller paragraphs, as with other results: otherwise it’s a LOT of information to try to digest.

Line 286: owners of – not the animals themselves

Line 287: to thermoregulation (as opposed to with)

Lines 289-90: reduced food intake were associated with many causes,

Line 290: hibernation OR pain

Line 302: difficulty coiling? (how was difficulty judged – absolutely inability vs ability to coil, or reduced coiling behaviour?)

Figure 2: hyporexia

Line 349: influence on??

Lines 377-380: restating what should be stated in the intro. The main opening should be what you found, eg confirmed or refuted each of your hypotheses.

Line 383: suggest replace “in turn” with “but”

Line 388-390: and presumably willing to seek veterinary attention for their animals.

Line 397: not sure what (…) signifies

Line 399: MBD is not just life threatening, it negativity impacts QOL eg mobility issues, reducing ability to perform normal behaviour

Line 406: replace Therefore likely with “it is possible…”

Lines 416-418: was it difficult to analyse because the data was not collected? In which case say the data we collected was limited to X, and therefore did not permit analysis of y…or if collected, what were the limitations. This sentence wasn’t so clear to me.

Line 434-436: This is where it would be useful to cite literature on online surveys to back up your claims.

Line 445: replace “are” with “were”

Line 460: It strikes me that it would be really useful in this paragraph to make cite Rowena Packer’s work on brachycephalic dogs re the consequences for welfare if owners perceive that BOAS is normal for the breed: Packer, R.M.; Hendricks, A.; Burn, C.C. Do dog owners perceive the clinical signs related to conformational

inherited disorders as ‘normal’ for the breed? A potential constraint to improving canine welfare. Anim. Welf.

2012, 21, 81–93.

Line 464-466: suggest reword, I think the point you are making here could be clarified.

Line 469-470: but what you are describing here seems to me denial, NOT a heuristic. Or at best a search satisfying bias.

Line 471: be careful. What were your aims in this study? Were you looking at enrichment and animal interactions with this? I just feel you don’t have the data to make this strong claim. Philosophically I am not at odds with you – I like making strong welfare claims, but if its too strong and your critics can prove you don’t have a basis to make it, you actually do harm to your cause (reptile welfare). Better to state the findings more modestly.

Line 478: high maintenance relative to?

Line 481: suggest replace “even” with “are likely”

Lines 485 - 487: why was this? And what should we do with this information?

Line 489: reference for where this figure came from?

Line 492-494: your results do not suggest this! Be careful not to overstate. Its okay that they don’t. You can suggest it is possible that this may be effective, but animal welfare campaigns are tricky. There is a whole science of human behaviour change that deals with the gap between human attitudes and behaviour. There’s nothing wrong with you suggesting it may be helpful to run such a campaign, but don’t get trapped into saying that it is more likely to be effective.

Line 495-497: I think this could be slightly expanded on. I am a bit confused as to how self-reported welfare assessments might be more helpful – would such not be subject to social desirability bias?

Line 529-531: care not to overstate. Also, don’t assume all reptile owners are well intended towards reptiles. Some may be keeping the animals for instrumental purposes.

Line 533: as above I didn’t agree with the heuristics approach. Happy to disagree but I think sometimes its just a matter of ignorance and/or denial.
